# Convergence of Neuroinflammation, Microbiota, and Parkinson’s Disease: Therapeutic Insights and Prospects

**DOI:** 10.3390/ijms252111629

**Published:** 2024-10-29

**Authors:** Nerea Domínguez Rojo, Mercedes Blanco Benítez, Ramón Cava, José Manuel Fuentes, Saray Canales Cortés, Rosa Ana González Polo

**Affiliations:** 1Departamento de Bioquímica y Biología Molecular y Genética, Facultad de Enfermería y Terapia Ocupacional, Universidad de Extremadura, 10003 Cáceres, Spain; nedominguezr@unex.es (N.D.R.); mercedesbb@unex.es (M.B.B.); jmfuentes@unex.es (J.M.F.); 2Instituto Universitario de Investigación Biosanitaria de Extremadura (INUBE), 10003 Cáceres, Spain; 3Centro de Investigación Biomédica en Red en Enfermedades Neurodegenerativas-Instituto de Salud Carlos III (CIBER-CIBERNED-ISCIII), 28029 Madrid, Spain; 4Tradinnoval Research Group, INBIO G+C, Departamento de Producción Animal y Ciencia de los Alimentos, Universidad de Extremadura, 10003 Cáceres, Spain; rcava@unex.es

**Keywords:** microbiota, neuroinflammation, Parkinson’s disease, therapeutic interventions

## Abstract

Parkinson’s disease (PD) is a complex neurodegenerative disorder. Recent evidence reveals connections between neuroinflammatory processes and intestinal microbiota alterations in the progression of this pathology. This comprehensive review explores the intricate relationships between them, highlighting their combined impact on PD. Neuroinflammation, characterized by immune activation in the central nervous system, is increasingly acknowledged as a critical factor in the development of PD. Concurrently, alterations in the gut microbiota composition have been linked to PD, suggesting a potential modulatory role in disease progression. Thus, bidirectional communication along the gut–brain axis has become pivotal in comprehending the pathogenesis of PD. Furthermore, we explore emerging therapeutic strategies that target these interconnected pathways, providing insights into potential avenues for PD treatment. The elucidation of these intricate relationships establishes a promising foundation for innovative therapeutic strategies aimed at altering disease progression and improving the quality of life for individuals affected by PD.

## 1. Introduction

Parkinson’s disease (PD) remains an intricate and multifaceted neurodegenerative disorder characterized by the selective degeneration of dopaminergic neurons within the substantia nigra, leading to a myriad of motor and non-motor symptoms [1]. Over the years, our understanding of PD pathogenesis has undergone a significant transformation, with growing recognition of the interconnected roles played by neuroinflammation and the gut microbiota in influencing disease progression [2].

### 1.1. Neuroinflammation in Parkinson’s Disease

Neuroinflammation, a prominent feature in PD pathology, involves the activation of microglia and the release of proinflammatory mediators within the central nervous system [3]. This inflammatory cascade contributes to the progressive loss of dopaminergic neurons, exacerbating motor deficits and contributing to the formation of Lewy bodies, the pathological hallmark of PD [4]. Emerging research continues to unveil the intricate molecular mechanisms underlying neuroinflammation in PD, offering novel insights into potential therapeutic targets [1].

### 1.2. The Gut Microbiota and the Gut–Brain Axis

Concomitantly, the gut microbiota, a diverse community of microorganisms inhabiting the gastrointestinal tract, has become a significant focus of PD research [5]. The bidirectional communication along the microbiota–gut–brain axis (MGBA) has become a key area of investigation, with evidence suggesting that the gut microbiota can influence brain function and, importantly, may play a role in neurodegenerative diseases such as PD [6]. Recent studies have identified alterations in the composition and diversity of the gut microbiota in PD patients, providing a basis for exploring the potential links between gut dysbiosis and disease pathology [7].

### 1.3. Interconnections and Emerging Insights

The gut–brain axis refers to the bidirectional communication network between the central nervous system (CNS) and the gastrointestinal tract, primarily mediated by the vagus nerve, the immune system, and the gut microbiota. This axis operates through neural, endocrine, and immune pathways. The gut microbiota plays a pivotal role in modulating immune responses by interacting with intestinal epithelial cells and immune cells, which triggers the release of cytokines and other signaling molecules. These immune mediators can cross the blood–brain barrier or influence brain function indirectly through neural pathways. Dysbiosis, or an imbalance in the gut microbiota, has been linked to neuroinflammation and altered CNS function. This suggests that microbial metabolites, such as short-chain fatty acids, can promote anti-inflammatory effects and regulate neuroinflammation. This immune modulation underscores the critical role of the gut microbiota in maintaining neuroimmune homeostasis, which impacts mental health and neurological disease [8,9].

The convergence of neuroinflammation and gut microbiota alterations presents an intriguing area of exploration. The intricate interplay between these factors has far-reaching implications for the pathogenesis and progression of PD [10]. Understanding how the gut microbiota influences neuroinflammatory processes and vice versa has become pivotal in deciphering the complex landscape of PD etiology.

This comprehensive literature review aims to synthesize the latest findings on the interconnections between neuroinflammation, the gut microbiota, and PD. By incorporating recent studies and advancements, we aim to provide an up-to-date and in-depth understanding of the dynamic relationships shaping PD pathology. Furthermore, we will explore emerging therapeutic strategies targeting these interconnected pathways, offering approaches into promising avenues for PD treatment.

### 1.4. Methods of Inducing Parkinson’s Disease

The induction of structural or functional changes in nigrostriatal dopaminergic transmission has been used to develop animal models of PD. The neurotoxins 1-methyl-4-phenyl-1,2,3,6-tetrahydropyridine (MPTP) and 6-Hydroxydopamine (6-OHDA) and the pesticide rotenone are the most used PD inducers [11]. Recently, the gram-negative *Bacillus Proteus mirabilis* has also been used in a study to induce the disease in a murine model [12]. All these agents simulate PD by inducing the loss of dopaminergic (DA) neurons in the substantia *nigra pars compacta*, together with a decrease in striatal dopamine levels and gastrointestinal dysfunction. Moreover, *Proteus mirabilis* is capable of inducing aggregates of α-synuclein, one of the main hallmarks of PD.

## 2. Types of Intervention

### 2.1. Diet and Dietary Supplementation

Nutrition and diet are now recognized as modifiable risk factors for chronic diseases. In recent years, studies have focused on these factors as potential avenues for reducing the risk of PD or slowing its progression [13]. Several studies have linked the consumption of fresh vegetables and fruits, nuts and other seeds, fish, herbs, and spices, with a reduced risk of PD progression or a lower incidence of PD. Conversely, certain foods, such as dairy products, have been associated with an increased risk of PD progression [14]. Consequently, it is imperative to investigate the diverse diets and their constituent elements to ascertain their impact on the various pathways associated with the initiation and progression of the disease.

Dietary habits significantly influence the intestinal microbiota, modifying its composition and contributing to alterations in microbiome diversity [15]. Furthermore, natural phytochemicals in foods have been shown to possess beneficial properties, including neuroprotective, anti-inflammatory, and antioxidant effects [16,17].

Recently, it has been suggested that changes in the microbiota resulting from dietary habits may be associated with the development of certain diseases, particularly those characterized by chronic inflammation, as observed in PD [18]. Consequently, it is of paramount importance to investigate the potential of diverse dietary patterns or nutritional supplements in modulating the intestinal microbiota and/or alleviating both motor dysfunction and inflammation in disease models (see Table 1 and Table 2).

The results of studies conducted by Zhang et al. [19] indicate that a ketogenic diet dominated by medium-chain triglycerides (MCT-KD) has a beneficial impact on the behavior of mice with PD. Mice under an MCT-KD exhibited increases in grip strength, distance run, and the time spent on the rotarod test (RTR) and a decrease in the time spent on the pole-climbing test (PT). The MCT-KD protected dopaminergic cells by increasing the number of TH^+^ cells in the substantia nigra pars compacta (SNpc) and elevated the levels of the neurotransmitter dopamine (DA), its metabolite 3,4-dihydroxyphenylacetic acid (DOPAC), and the dopamine transporter DAT. Furthermore, it reversed the effects of MPTP on the nervous system. Additionally, the MCT-KD reversed dopaminergic cell death in PD mice, resulting in an increase in the expression of the anti-apoptotic protein Bcl-2 and a concomitant decrease in the expression of the pro-apoptotic protein Bax. In contrast, the MCT-KD was found to reduce the levels of the proinflammatory cytokines, such as tumor necrosis factor-α (TNF-α), interleukin-1β (IL-1β), and interleukin-6 (IL-6) in the SNpc, which have been demonstrated to increase in response to MPTP. The MCT-KD mitigates the dysbiosis induced by MPTP, reducing the abundances of *Aminobacterium*, *Desulfomicrobium*, *Fermentimonsa*, *Fibrobacter*, and *Ruminiclostridium* while increasing the relative abundances of *Blautia*, *Mycobacterium*, and *Ruminiclostridium*. The MCT-KD not only alters the composition of the gut microbiota but also affects the metabolic phenotype in the nervous system (NS), increasing metabolites such as anserine, perfluorohexanoic acid, D-pantothenic acid, and the chemical salt of lactamide while decreasing deoxycholylcitrulline, adlupulone, linoleic acid-d4, ganoderiol I, lysoPC, and glabrene [19].

Another dietary approach involves the administration of high levels of branched-chain amino acids (BCAAs). In a rotenone-induced mouse PD model, Xhenxhen et al. observed improvements in motor behavior and neuronal protection, as well as an increase in the number of TH^+^ cells. Consequently, this dietary regimen mitigates both NS and colon inflammation, concurrently reducing the levels of the proinflammatory cytokines TNF-α, IL-1β, and IL-6 [21]. Similarly, Na et al. employed this diet in a model of MPTP-induced PD, demonstrating comparable behavioral outcomes. The researchers also found that this dietary regimen increased the frequency, weight, and water content of the feces in mice with PD. Additionally, differences in the short-chain fatty acid (SCFA) content were observed, with a notable increase in propionic acid levels and a significant decrease in isovaleric acid levels [20].

Similarly, [22] revealed that a fasting-mimicking diet (FMD) can enhance motor function and mitigate the loss of dopaminergic neurons in the SNpc resulting from MPTP exposure. Furthermore, an FMD has been observed to elevate the levels of brain-derived neurotrophic factor (BDNF), a neuroprotective factor associated with the survival of dopaminergic neurons. Conversely, the FMD demonstrated a reduction in the number of Iba-1^+^ cells and the release of TNF-α and IL-1β, indicating its capacity to suppress neuroinflammatory processes.

Sun et al. [17], using the A53T transgenic mouse model, demonstrate significant motor deficits, which were reversed in a concentration-dependent manner using resveratrol (RES). Additionally, these changes were more pronounced when RES was encapsulated with hydroxy-propyl beta-cyclodextrin (RHSD). The gastrointestinal dysfunction observed in PD mice showed improvement following treatment with RES-L, RES-H, and RHSD, as evidenced by the increased fecal water content and fecal output, with RHSD demonstrating the most pronounced effect. RHSD enhanced the expression of ZO-1 and occludin, reinforcing intestinal barrier integrity. The administration of RHSD and RES-H reduced α-syn accumulation in both the colon and brain of A53T mice. Furthermore, RHSD has been observed to reduce the number of Iba-1-positive cells in the substantia nigra. The administration of RHSD to PD mice established a microbiota composition in PD mice comparable to that observed in wild-type mice, effectively reversing the dysbiosis induced by the pathology. Additionally, there was a notable increase in intestinal probiotics, specifically *Lactobacillus murinus* and *Lactobacillus reteri*. A53T mice exhibited an increased abundance of microbiota associated with inflammation and motor deficits, along with elevated levels of metabolites such as lithodpermoside, 5-(Diphenylphosphinyl) pentanoic acid, sulfadimethaxine, and vinblastine [17]. In the same model, Lu et al. [34] demonstrated that baichanting (BCT) yielded similar outcomes; BCT also reduced oxidative stress, leading to increased DA and superoxide dismutase (SOD) release, along with reductions in the proinflammatory factors TNF-α, IFN-β. IL-1β, IL-6, nitric oxide (NO), and malondialdehyde (MDA). Notably, BCT regulated the abundance of some microbial genera, thereby restoring intestinal microbiota function and reversing fecal metabolic abnormalities.

Avagliano et al. [31] showed that 6-OHDA produces a persistent motor deficit. This fact is manifested in the apomorphine test, where it induces an increase in the number of total rotations and a depletion of DA in the PD model. In contrast, repeated administration of BuNa (butyrate) ameliorates the described motor deficits by reducing apomorphine-induced contralateral rotations. This intervention also alleviates the behavior in the model mice, associated with reductions in several pathogenic factors in the striatum. At the peripheral level, BuNa reduces systemic inflammation, further counteracting colonic morphological changes related to 6-OHDA treatment, and remodels the intestinal metataxonomics, possibly triggered by intestinal mucin synthesis allowing the growth of *Akkermansia muciniphila*. Thus, we can conclude that this mechanism could be responsible for the improvement in intestinal integrity and intestinal barrier function.

In another study [30], a rotenone-induced disease model highlighted the augmented weight loss and motor dysfunction, which were alleviated by a derivative of schamosamide (FLZ). FLZ improved the intestinal transit reduction and the reduction in colon length, present in the model, at week 6, a week in which FLZ significantly increased the percentage of water content and fecal pellet production, as it does at week 5. Consistent with these results, rotenone induction reduced TH^+^ cells and increased the number of Iba-1^+^ and GFAP^+^ cells. Again, FLZ reversed these effects, particularly reversing the reduction in TH^+^ cells, and reduced GFAP^+^ astrocytes and Iba1 mRNA expression in rotenone-induced mice. Furthermore, FLZ decreased the levels of the neuroinflammatory markers Cd3, IL-1β, IL-6, Cox2, and NOS2 in NS, which were increased by rotenone.

Huh et al. used a mouse model treated with *P. mirabilis* to induce motor deficits, which were significantly reversed by 6-shogaol (6S), a dehydrated form of 6-gingerol, and ginger extract (GE) treatment, especially GE. GE and 6S increased the number of TH^+^ cell counts and the DAT intensity in *P. mirabilis*-treated mouse brains, reduced the accumulation of a-syn filaments in the NS and colon, inhibited microglial activation induced by *P. mirabilis*, and reduced the number of Iba-1^+^ cells, thus protecting against brain and colon inflammation. In the colon, in addition to reducing the increase in TNF-α, 6S improves intestinal barrier integrity as observed by an increase in occludin [12].

He et al. reported that neohesperidin (Neo) reversed MPTP-induced weight loss and improved motor behavior in mice, increasing TH, DA, and DOPAC expression in the brain while attenuating the increases in Iba-1 and OX42. Neo reduced the proinflammatory cytokines IL-6, IL-1β, and TNF-α in both the brain and the colon; increased the alpha-diversity; and shifted the b-diversity composition to one that was similar to controls. Thus, Neo increased the levels of *Prevotella* and *Bacteroides* and decreased *Actinobacteria* and *Proteobacteria* in the MPTP mouse. Furthermore, Neo increased gastrointestinal barrier integrity by enhancing ZO-1 and Claudin-3 expression [23].

Similar results were observed with NaB [24], CDG [25], OFO [27], and GEP [26]. NaB increased serotonin (5-HT) and 5-HIAA and reduced cleaved caspase-3, while GEP and OFO increased the Bcl-2/Bax ratio, confirming the protection against dopaminergic neuron death. As for α-syn accumulation, it is inhibited by GEP.

Additionally, GEP, CDG, and NaB reduced the number of GFAP^+^ cells, while CDG decreased MDA and increased SOD activity, contributing to inflammation reduction induced by MPTP. OFO reduced the population of *Lachnospiraceae*_UCG-001, a key intestinal bacterium linked to cognitive impairment, and increased the major SCFAs such as acetic acid and butyric acid. OFO also enhanced claudin-1, occludin, and Muc2 levels in the colon. GEP elevated occludin levels and SCFA levels. Furthermore, NALL demonstrated improvements in motor behavior, increased rotarod performance, and elevated TH levels [28].

Curcumin administration in the MPTP-induced PD model [35] prevented motor impairment and dopaminergic neuronal loss and significantly reduced glial cell activation in the SNpc and striatum. On the other hand, chicoric acid (CA) [32] and maslinic acid (MA) [33] produced similar effects on motor behavior, neuroinflammation, and dopaminergic cell death protection, while there was a lower loss observed for BDNF and GDNF. However, CA and MA produced opposite results on the intestinal microbiota, because while AC increased *Bacteroidetes* and decreased *Firmicutes*, MA at low doses had opposite effects on the intestinal microbiota, specifically increasing certain *Firmicutes* families, such as the *Lactobacillaceae* family, or decreasing the abundance of others, such as the *Ruminococcaceae* or *Rikenellaceae* family. The results with AC could modify the microbial imbalance present in the PD model, producing a modulation of both the diversity and the composition of the intestinal microbiota and modulating SCFAs and potentially restoring SCFA production.

In a PD model induced by MPTP and LPSm [29], Hua-Feng-Dan (HFD) and 70W treatments increased the rotarod duration and improved motor deficits, increased TH-IR+ cells, and reduced activated microglia. HFD and 70W decreased the *Verrucomicrobiota_Verrucomicrobaceae* abundance, characteristic of PD models [36].

### 2.2. Probiotics

The mounting evidence indicates that the gut microbiota plays an important role in the pathology of Parkinson’s disease (PD) [37]. Furthermore, it suggests that the modulation of the MGBA may represent a promising therapeutic approach for ameliorating motor syndromes and preventing disease progression [38].

Probiotics, defined as live microorganisms that confer a health benefit on the host when administered in adequate amounts, represent a promising class of such modulators [39]. A growing body of evidence from preclinical studies indicates that specific probiotic strains can improve behavioral outcomes, markers of neuroinflammation, and microbiota dysbiosis in PD models [40]. More specifically, *Lactobacillus* and *Bifidobacterium* species have been successfully used in established animal models of PD, both individually and in combination. However, the specific probiotic strains used and the conditions in the host are likely to influence the clinical results observed [41].

Table 3 presents the findings of recent studies conducted on murine models of PD induced by different types of toxins, investigating the impact of distinct probiotics.

Jian-Fu Liao et al. [37] demonstrated that the administration of *Lactobacillus plantarum* PS128 (PS128) in MPTP-treated mice significantly alleviated motor deficits in the PT, narrow-beam test (NBT), and RTR. Furthermore, the behavioral improvement correlated with increased DA levels and significant improvements in metabolite levels, including DOPAC and HVA. PS128 treatment also increased the number of TH^+^ neurons in the NSs of MPTP-treated mice, along with TH protein expression levels. PS128 treatment significantly reduced glial activation induced by MPTP, as evidenced by the decreased expression levels of GFAP and Iba1 proteins and reduced the levels of the proinflammatory cytokines TNF-α, IL-1β, and IL-6. This study also explored the impact of PS128 on the gut microbiota modulation, revealing a reduction in the *Firmicutes abundance* and an increase in the *Bacteroidetes* abundance compared to PD mice.

Interestingly, Yan Zhang Lee et al. [42] reported similar results in a rotenone-induce PD mouse model. The authors found that PS128 regulates the expression of certain miRNAs that contribute to the Parkinson’s pathogenesis via microglial activation. Specifically, PS128 administration significantly reduced the levels of miR-155-5p and miR-223-3p compared to PD mice. Consequently, the levels of miR-155-5p correlated with the increases in the expression of its direct target, the suppressor of cytokine signaling 1 (SOCS1) protein and Socs1 mRNA. In addition, PS128 modulated the gut microbiota in PD mice, enriching the relative abundances of *Bifidobacterium*, *Ruminiclostridium*_6, *Adlercreutzia*, *and Acetifactor* compared to PD mice while the abundances of *Ruminococcaceae*_UCG_014, *Bacteroides*, and *Alistipes* were decreased. These results were positively associated with improvements in rotenone-induced motor deficits, and the abundances of *Bifidobacterium*, *Ruminiclostridium*_6, *Bacteroides*, and *Alistipes* were correlated with the miR-155-5p/SOCS1 pathway in PD mice.

Additionally, the therapeutic potential of other strains of *Lactobacillus plantarum* as probiotics has been explored in murine PD models. Lei Wang et al. [39] investigated the therapeutic potential of *Lactobacillus plantarum* DP189 (DP189) in PD mice via MPTP. DP189 reduced α-synuclein expression and the expression levels of inflammatory factors in the SN, including TNF-α, IL-1β, IL-6, and NLRP3, in MPTP-treated mice. Furthermore, DP189 treatment modulated the composition of the gut microbiota, increasing the abundances of *Prevotella*, *Clostridium*, *Bacteroides*, and *Lactobacillus* while reducing harmful bacteria such as *Leptospira*, *Acidovorax*, *Aquabacterium*, *Candidatus arthromitus*, and *Novosphingobium* in PD mice. Similar results were observed with *Lactobacillus plantarum* CCFM405 in a rotenone-induced PD model [43] where this probiotic enhanced branched-chain amino acid synthesis, including nicotinic acid, L-valine, L-isoleucine, and L-leucine in feces.

*Bifidobacterium* species have also been successfully used in well-stablished PD models. The administration of *Bifidobacterium animalis subs. Lactis* NJ241 (NJ241) in an MPTP-induced mouse model [44] significantly increased locomotor activity, rearing behavior, and wire hanging ability compared to MPTP mice. Furthermore, NJ241 intervention demonstrated effects similar to previous studies, including reduced damage to dopaminergic neurons and glial activation in the SN. These results correlated with an increase in the expression of ZO-1 and occludin in relation to MPTP-induced mice. In this study, Yuxuag Dong et al. also demonstrated that NJ241 modulated the gut microbiota and elevated SCFA levels in PD mice, which correlated with the elevated colonic levels of intestinally derived hormone (GLP-1) and its receptor (GLP-1R) compared to MPTP mice. Additionally, NJ241 ingestion increased PGC1 expression, a transcriptional coactivator involved in the regulation of cellular metabolism and energy homeostasis [44]. These results suggest that NJ241 can modulate the inflammatory response through the GLP-1R/PGC1 signaling pathway.

Interestingly, supplementation with the probiotic *Bifidobacterium breve* CCFM1067 [41] in an MPTP-induced mouse model of PD resulted in behavioral improvement, which correlated with elevated neurotransmitters and certain metabolite levels. In addition, similar to other probiotics, CCFM1067 reduced the expression of the glial markers Iba-1 and GFAP and increased the expression of the neurotrophic factors BDNF and GDNF. Another consequence of *B. breve* CCFM1067 was a reduction in proinflammatory cytokines and an increase in IL-10, TNF-α, IL-1β, and IL-6 levels. Moreover, *B. breve* CCFM1067 treatment has been demonstrated to effectively modulate the alterations observed in the gut microbiota observed in MPTP-treated mice, in particular, increasing the relative abundance of the beneficial bacteria *Akkermansia* and reducing harmful bacteria such as *Bacteroidetes*, *Escherichia-Shigella*, and *Dubosiella.*

The scientific literature suggests that several probiotics, when used in symbiotic combinations, exert effects against the pathogenesis of PD. Marco Sancandi et al. [38] have demonstrated that Symprove^TM^, a probiotic suspension containing four bacterial strains—*Lactobacillus acidophus* NCIMB 30175, *Lactobacillus plantarum* NCIMB 30173, *Lactobacillus rhamnosus* NCIMB 30174, and *Enterococcus faecium* NCIMB 30176—was shown to significantly influence both the gut and brain in a PD mouse model induced with 6-OHDA. Specifically, Symprove^TM^ supplementation significantly decreased inflammatory markers in the plasma, including TNF-α, IL-1β, and IL-6, compared to PD mice. Furthermore, this probiotic increased the α-diversity and modified the β-diversity in PD mice.

### 2.3. Fecal Microbiota Transplantation

Fecal microbiota transplantation (FMT) is a procedure in which the fecal microbiota from a healthy individual is transferred to a patient with a specific disease. The fecal microbiota is extracted from processed stool samples and transferred to the patient’s intestinal tract [45,46]. This therapy is currently gaining traction due to its ability to promptly regulate imbalances in the gut microbiota compared to probiotics and prebiotics [47].

Despite the observed dysbiosis of the gut microbiota and neuroinflammation in PD, there has been limited comprehensive investigation into the efficacy of FMT. However, studies focusing on this specific form of therapy are gradually becoming more prevalent. Table 4 summarizes the studies analyzing the beneficial effects of this therapy in different models of PD.

In this context, Zhao et al. [48] demonstrated that FMT alleviates motor symptoms and gastrointestinal dysfunction in the rotenone-induced PD mouse model. In the NS of rotenone-treated mice, there was a significant reduction in neurons, to nearly half the level in controls. However, FMT was able to restore this loss.

Furthermore, the researchers demonstrated that FMT markedly attenuates the histological features (immune infiltration and epithelial deterioration in the colon associated with PD and inflammation) in both the SN and colon of rotenone-treated mice. Specifically, FMT administration inhibits the TLR4/MyD88/NF-κB signaling pathway associated with lipopolysaccharide (LPS) exposure.

In order to gain a deeper insight into the MGBA mechanisms in rotenone-induced PD pathogenesis and the protective effects of FMT, the researchers conducted correlation analyses between different microbiota populations and NS alteration parameters, including the rotarod duration, TH-positive neuron numbers in the SN, LPS levels, ZO-1, and TLR4 expression in the NS midbrain. Most correlations between these markers were statistically significant. These correlation results collectively indicate the involvement of the microbiota–gut–brain axis in the chronic rotenone mouse model pathogenesis and the protective effects of FMT in this model.

The MPTP-induced mouse model is one of the most used PD models. Several studies have examined the beneficial effects of FMT in this model [49,50,51]. Zhang et al. [49] demonstrated that FMT not only alleviates dyskinesia in PD mice but also improves the major microbial populations of MPTP-exposed mice, including *Anaerostipes*, *Bifidobacterium*, ASF356, and *Ruminococcus*. Additionally, this study indicated that FMT alters the expression levels of various inflammatory markers, such as arginase, GSK3β, IL-1β, iNOS, p-PTEN, and α7n AchR. However, further investigation is required to determine whether this process reduces TNF-α, IL-1β, and TGF-β pathway activities in the gut.

A further study [51] demonstrated that FMT from healthy human controls can correct intestinal dysbiosis and ameliorate neurodegeneration in the MPTP-induced PD mouse model. This is achieved by suppressing microgliosis and astrogliosis, improving mitochondrial alterations via the AMPK/SOD2 pathway, and restoring nigrostriatal pericyte loss and blood–brain barrier (BBB) integrity.

The results of the study by Zhang et al. [50] indicated that the aged microbiota response to MPTP treatment differed from that of the young microbiota. FMT from aged mouse donors partially resists motor dysfunction caused by MPTP, enhances reductions in the striatal 5-HT content, and slows down DA turnover.

Regarding the microbiota composition, no significant differences in α-diversity were observed, suggesting minimal structural differences between the experimental groups. However, β-diversity demonstrated notable alterations in the microbiota composition, with the abundances of *Akkermansia*, a potential next-generation probiotic linked to host mucus turnover, which enhances intestinal barrier function [50], and *Candidatus Saccharimonas* being decreased in MPTP-exposed mice transplanted with young microbiota, whereas these increased significantly in disease-transplanted mice with aged microbiota. This finding suggests the potential for the aged microbiota to play a protective role in the development of PD. However, FMT from aged donors does not appear to alleviate striatal neuroinflammation, induce phenotypic changes in striatal glia cells, or affect striatal cytokine expression in PD mice or influence colonic inflammation levels. These findings confirm that FMT from an aged host does not affect intestinal barrier permeability, or the intestinal inflammatory response caused by MPTP treatment. Ultimately, the aged microbiota appears to not influence the development of PD.

### 2.4. Others

In addition to the aforementioned interventions, we have included several other potential interventions under the category of “others”, including umbilical cord blood plasma and antibiotics, either alone or combined with FMT. Table 5 provides an overview of some of these studies.

#### 2.4.1. Umbilical Cord Blood Plasma Therapy

Umbilical cord blood plasma (UCBP) is the fluid that remains after the removal of red blood cells, white blood cells, and platelets from the umbilical cord blood and placenta. It contains a range of proteins, growth factors, hormones, and other molecules crucial for cell communication and tissue regeneration. Its accessibility and unique characteristics make it a valuable resource in various therapeutic applications. The growth factors and cytokines in UCBP stimulate cell proliferation and differentiation, aiding tissue regeneration after damage.

A plethora of studies have investigated the advantages of utilizing UCBP in different contexts, such as to treat chronic inflammatory conditions including inflammatory bowel disease, where its anti-inflammatory properties help reduce inflammation and improve bowel function [57]. Additionally, UCBP shows promise in treating neuropsychiatric disorders such as autism spectrum disorder by reducing inflammation and promoting neuronal repair [58]. In acute injuries such as liver injury and acute lung injury, UCBP accelerates the healing process and reduces scarring [59,60]. Its angiogenic properties suggest potential uses in the treatment of cardiovascular diseases, which may promote the formation of new blood vessels and enhance cardiac tissue repair [61].

Current studies reveal the potential for UCBP to be used in combination with other therapies, such as mesenchymal stem cells (MSCs) and gene therapies, to treat a variety of diseases and improve overall health [62,63]. Among these diseases are neurodegenerative disorders, in particular PD.

Table 5 presents two studies focused on the beneficial effects of UCBP in MPTP mouse models. Lee et al. [52] demonstrated that treatment with UCBP significantly improved motor and non-motor function in MPTP-treated animals. A reduction in the amount of dopaminergic cell (TH) loss in the SNpc and immune cell (MHCII) activation is described. TNF-α expression was also reduced in the brain and gut of toxin-exposed mice and significantly reduced inflammatory gut microbiota populations. Similarly, Sun et al. [53], using intranasal administration of human umbilical cord mesenchymal stem cells (UC-MSCs), corroborated Lee’s findings. In addition to improving motor dysfunction, decreasing the degeneration of dopaminergic neurons, and decreasing the release of proinflammatory cytokines, UC-MSC treatment decreased the relative abundance of *Escherichia-Shigella*, ameliorating the intestinal dysbiosis observed in MPTP-treated mice. *Escherichia* and *Shigella* have been shown to activate microglia through amyloid protein secretion, inducing oxidative stress and inflammatory cytokine release [64]. This process induces oxidative stress and the release of inflammatory factors, including TNF-α, IL-1β, and IL-6 [65]. UC-MSCs also attenuated Lewy bodies within the brain, consequently curbing the aberrant aggregation of α-Syn within the gut and further attenuating intestinal inflammation.

#### 2.4.2. Antibiotics

The search for compounds with protective properties is hindered by the difficulty of identifying suitable candidates, leading to the exploration of existing compounds for new functions.

Antibiotics, traditionally used to combat bacterial infections, have shown neuroprotective properties [66,67,68]. Long-term antibiotic use has been demonstrated to significantly alter the composition and diversity of the gut microbiota [69]. This effect has been associated with efficacy in reducing microglia and α-Syn protein accumulation, improving PD-associated motor deficits [70,71,72].

Table 5 outlines studies focused on the use of antibiotics as an intervention to ameliorate PD-associated disorders in the murine model exposed to MPTP. However, the prolonged use of antibiotics can lead to severe depletion of the gut microbiota, which can negatively affect different organ functions, both metabolic and immunological [70]. In this context, two of the three studies presented combine antibiotics with FMT to mitigate these adverse effects [54,55].

Zhou et al. [54] demonstrated that ceftriaxone, a broad-spectrum antibiotic of the third-generation cephalosporin class primarily used for severe and complicated bacterial infections [73], alleviated the behavioral and neuropathological changes induced by MPTP. Ceftriaxone reduced inflammation-associated protein expression, including TLR4, MyD88, and NF-κB, in both the colon and the brain and decreased serum proinflammatory cytokine levels such as IL-1β, IL-6, and TNF-α. Additionally, ceftriaxone lowered the levels of the genus Proteus pathogenic bacteria and increased the probiotic *Akkermansia*.

Mao et al.’s group [55] achieved similar results using the antibiotic dioscin, a bioactive steroidal sapogenin with excellent anti-inflammatory and antioxidant activities [74,75,76]. Dioscin improved the PD phenotype by restoring intestinal dysbiosis, regulating bile acid-mediated oxidative stress, and neuroinflammation via GLP-1 signaling in MPTP-induced PD mice. Dioscin increased the relative abundances of Clostridiales and *Enterobacteriales* at the family level and reduced the relative levels of *Enterococcus*, *Streptococcus*, *Bacteroides*, and *Lactobacillus* at the genus level. This further inhibited bile salt hydrolase (BSH) activity and blocked bile acid (BA) deconjugation. The FMT test demonstrated that the anti-PD effect of dioscin was dependent on the gut microbiota. Furthermore, the experiment used ursodeoxycholic acid (UDCA), a natural bile acid used in treating liver diseases such as primary biliary cholangitis and gallstones, to enhance dioscin’s protective effects [77].

It is also worth noting that vancomycin improved motor dysfunction in mice exposed to MPTP, as demonstrated in a study by Cui et al. [56]. Vancomycin was selected for its ability to selectively eliminate gram-positive bacteria. Despite the lack of effect on DA or the DA synthesis process, vancomycin inhibited DA metabolism by suppressing the expression of striatal monoamine oxidase B (MAO-B), a key enzyme in DA metabolism [78]. Vancomycin markedly reduced the number of astrocytes and microglia, and microglia activation was significantly reduced in the substantia nigra pars compacta (SNpc) of mice with PD that received vancomycin pretreatment. These results indicate that vancomycin pretreatment can suppress glial activation in MPTP-induced PD mice, probably mediated by a reduction in the MAO-B level. The diversity analysis focused on the abundance and distribution of the gut microbiota reported that the genus *Akkermansia* exhibited the most pronounced upregulation following vancomycin pretreatment, linked to a negative correlation with proinflammatory cytokines and chemokines in several animal disease models [79].

Figure 1 illustrates the principal pathophysiological alterations observed in the investigated murine models of PD and the restoration achieved through the analyzed interventions.

Additionally, Figure 2 depicts a comparison of the predominant microbiota genera that are altered or unaltered in PD in response to therapeutic interventions.

## 3. Human Interventions

In contrast to the extensive interventions conducted in animal models, particularly in mice, there are limited data on interventions aimed at modifying the gut microbiota in PD patients. Notably, the Mediterranean diet has been shown to influence the gut microbiota composition in a cohort of Parkinson’s patients compared to control individuals. After a five-week adherence period, improvements in constipation symptoms and modulation of the microbiota were observed in individuals with PD; however, no significant changes in intestinal permeability or motor symptoms were reported [80].

In another study, Tamtaji et al. [81] conducted a randomized, double-blind, placebo-controlled trial assessing the effects of probiotic supplementation (Lactobacillus acidophilus, Bifidobacterium bifidum, Lactobacillus reuteri, and Lactobacillus fermentum, each at 2 × 10^9^ CFU/g) over 12 weeks in individuals with PD. The results indicated a modest impact on the MDS-UPDRS scale, although the small sample size necessitates caution and further population studies.

Fecal microbiota transplantation (FMT) is perhaps the intervention with the most available data in humans, although the results have been divergent. For instance, Segal et al. [82] reported significant improvements in both motor and non-motor symptoms in a group of six patients at 24 weeks post-intervention. This finding is consistent with Cheng et al. [83], which showed similar results in a larger cohort of 56 Parkinson’s patients after a 12-week oral treatment. However, the small sample sizes in both studies remain a significant limitation. In contrast, a very recent study [84] involving a larger cohort found no significant changes in motor and non-motor symptoms.

In conclusion, both dietary interventions (with or without probiotics) and fecal microbiota transplantation represent promising areas of research concerning the relationship between the gut microbiota and the development and progression of Parkinson’s disease (Table 6).

## 4. Conclusions

This comprehensive review of dietary interventions, probiotic treatments, FMT, umbilical cord blood plasma therapy, and antibiotics underscores the diverse approaches currently under investigation for managing PD via microbiome modulation. Dietary modifications, particularly those emphasizing ketogenic diets, branched-chain amino acids, fasting-mimicking diets, and specific natural compounds such as resveratrol and curcumin, have demonstrated beneficial effects in mitigating PD symptoms. These benefits are achieved through mechanisms involving neuroprotection, anti-inflammation, and gut microbiota modulation in various animal models.

Probiotic interventions, especially those utilizing specific strains of Lactobacillus and Bifidobacterium, show promise in correcting dysbiosis, reducing neuroinflammation, and improving motor functions in PD models. The impact of probiotics on the gut–brain axis, as evidenced by numerous studies, underscores their therapeutic potential.

FMT has emerged as a promising intervention for PD treatment, with studies demonstrating its ability to alleviate motor symptoms and gastrointestinal dysfunction by restoring the gut microbiota composition. The capacity of FMT to modulate neuroinflammation and enhance neuronal survival further supports its potential as a PD treatment strategy.

Additionally, novel interventions such as umbilical cord blood plasma therapy and antibiotic treatments, either alone or combined with FMT, are being explored for their therapeutic benefits in PD. Umbilical cord blood plasma therapy presents a novel approach, with studies indicating its effectiveness in improving motor and non-motor functions, reducing neuroinflammation, and modulating the gut microbiota in PD models. However, further clinical research is needed to fully elucidate the long-term efficacy and safety of this approach.

Antibiotics, specifically in combination with FMT, offer another avenue for addressing PD-associated dysbiosis and inflammation. While antibiotics such as ceftriaxone and vancomycin have shown efficacy in preclinical models, their long-term impact on the gut microbiota and overall health necessitates careful consideration.

Collectively, these diverse interventions underscore the importance of targeting both central and peripheral factors in PD pathogenesis. The interplay between diet, the microbiota, and neuroinflammation presents a compelling framework for developing comprehensive treatment strategies. Future research should focus on optimizing these interventions, understanding their long-term effects, and exploring their potential synergistic benefits in clinical settings.

## Figures and Tables

**Figure 1 ijms-25-11629-f001:**
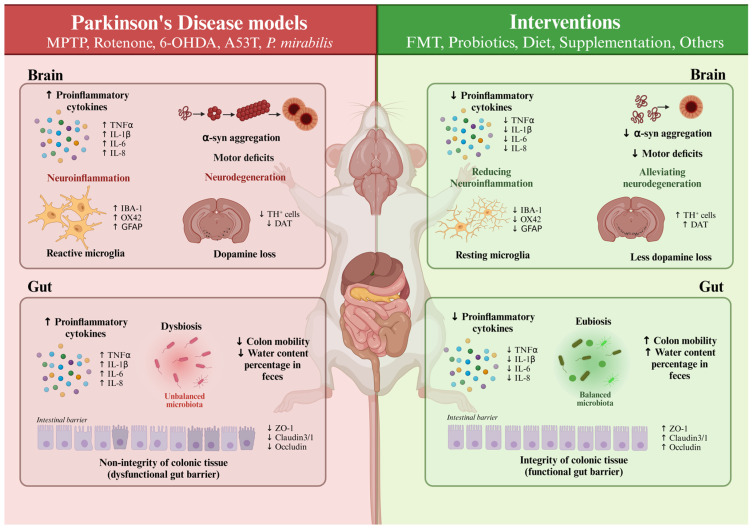
Comparison of pathophysiological features between Parkinson’s disease (PD) murine models and the different interventions studied. TNF-α: Tumor Necrosis Factor α; IL-1β: Interleukin-1β; IL-6: Interleukin-6; IL-8: Interleukin-8; Iba-1: Ionized Calcium-binding Adaptor Molecule 1; GFAP: Glial Fibrillary Acidic Protein; α-syn: α-Synuclein; TH: Tyrosine Hydroxylase; DAT: Dopamine Transporter.

**Figure 2 ijms-25-11629-f002:**
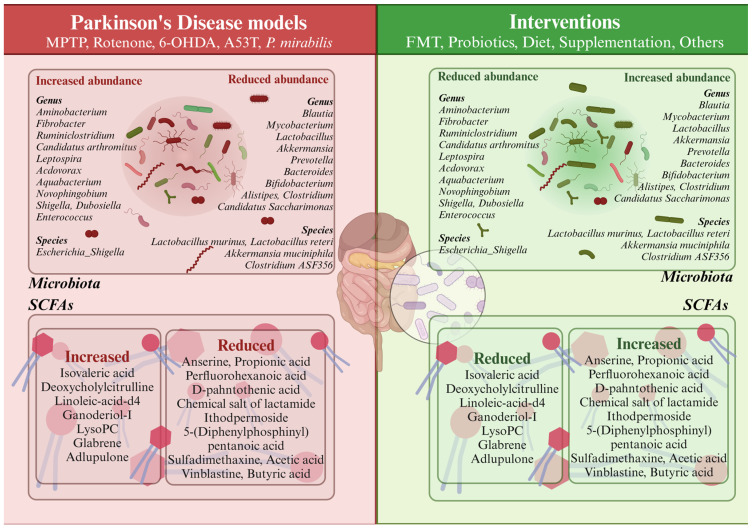
Comparison of the microbiota and short-chain fatty acids (SCFAs) between Parkinson’s disease (PD) murine models and the different interventions studied.

**Table 1 ijms-25-11629-t001:** Dietary intervention studies. BCAA: Branched-chain Amino Acids. DA: dopamine. FMD: Fasting-mimicking Diet. MPTP: 1-Methyl-4-phenil-1,2,3,6-tetrahydropyridine.

	PD Model					Intervention			
Author	Model	Toxic	Dose	Duration	Administration	Diet	Duration	Administration	Results
[19]	C57BL6 mice	MPTP	25 mg/kg	35 days	Intraperitoneal	Ketone diet (MCT-KD)	42 days	Oral	Protection against the loss of DA neurons and alterations in the gut microbiota and its metabolites.
[20]				7 days		High-BCAAs (H-BCAA)	56 days		Relief of both motor and non-motor symptoms, promoting SCFAs.
[21]		Rotenone	30 mg/kg	21 days/28 days	Gavage		360 days		Anti-inflammatory effect in the gut and brain
[22]		MPTP	20 mg/kg		Intraperitoneal	Fasting-mimicking diet (FMD)	Three 7-day cycles (FMD for 3 consecutive days in each cycle)		Recovery of motor function, mitigation of loss of dopaminergic neurons, and increase in DA and 5-HT levels.

**Table 2 ijms-25-11629-t002:** Dietary supplementation intervention studies. MPTP: 1-Methyl-4-phenil-1,2,3,6-tetrahydropyridine. FLZ: N-[2-(4-Hydroxy-phenyl)-ethyl]-2-(2,5-dimethoxy-phenyl)-3-(3-methoxy-4-hydroxy-phenyl)-acrylamide. RHSD: Resveratrol/HP-b-CD inclusion complex.

	PD Model						Intervention				
Author	Model	Toxic	Dose	Duration	Administration		Compound	Dose	Duration	Administration	Results
[23]	C56BL/6J mice	MPTP	25 mg/kg	7 days	Intraperitoneal		Neohesperidin	25 mg/kg–50 mg/kg	24 days	Oral (gavage)	Attenuation of neurodegeneration by increasing the population of beneficial intestinal bacteria.
[17]	A53T transgenic mice						Resveratrol/RHSD	30 mg/kg–90 mg/kg RES or 30 mg/kg RHSD	30 days once every 3 days	Oral (gavage)	Reversion of gut microbiota dysbiosis. Improvement in motor dysfunction and pathological progression.
[24]	C57BL/6 mice	MPTP	30 mg/kg	7 days	Intraperitoneal		Sodium butyrate	300 mg/kg/day	21 days	Intragastric	Restauration of the gut microbiota composition and inflammation.
[25]	C57BL/6 mice	MPTP	30 mg/kg	5 days	Intraperitoneal		Compound Dihuang Granule	10 g/kg	7 days	Oral	Regulation of gut microbiota dysbiosis, reduction in neuroinflammation and induction of neuroprotective effects.
[12]	C57BL/6J mice	Proteus Mirabilis (*P. mirabilis*)		2 × 10^8^ CFU/0.2 mL sterilized PBS	5 days	Oral	Ginger 6-Shogaol	300 mg/kg/day GE 10 mg/kg/day 6S	22 days	Oral	Mitigation of *P. mirabilis*-induced motor behaviors and dopaminergic neuron degeneration through the regulation of the gut microbiome.
[26]	C57BL/6 mice	MPTP	30 mg/kg/day	7 days	Intraperitoneal		Gastrodia elata polysaccharide	600, 400, or 200 mg/kg/day	21 days	Oral	Mitigation of motor dysfunction, inhibition of α-syn aggregation, and attenuation of DA loss. Alleviation of neuroinflammation through the regulation of the gut microbiota and increased secretion of SCFAs.
[27]	C57BL/6J mice	MPTP	25 mg/kg	5 days	Intraperitoneal		Oral [60] fullerene	0.5655 mg/kg, 1.131 mg/kg, and 2.262 mg/kg	12 days	Oral	Prevention of dopaminergic neurodegeneration by reducing neuroinflammation. Improvement in motor dysfunction and reversal of DA loss through the regulation of the gut microbiome.
[28]	C57BL/6 mice	MPTP	30 mg/kg	5 days	Intraperitoneal		N-acetyl-L-leucine	100 mg/kg/day	7 days	Oral	Neuroprotective effect on DA neurons by regulating the gut microbiota.
[29]	C57BL/6 mice	MPTP + LPS	15 mg/kg + 5 mg/kg	4 days	Intraperitoneal		Hua-Feng-Dan or 70W	0.6 g/kg or 0.2 g/kg	35 days	Oral	Protection against neurotoxicity by modulation of the gut microbiome.
[30]	C57BL/6J mice	Rotenone	30 mg/kg/day	28 days	Oral		FLZ	75 mg/kg/day	14 days	Oral	Alleviation of gastrointestinal dysfunctions and motor deficits by attenuation of microbiota dysbiosis.
[31]	Swiss CD1 mice	6-OHDA	4 µg/2 µL	14 days	Intrastriatal injection		Butyrate	100 mg/kg	14 days	Oral	Improvement in motor deficits associated with reduced expression of inflammatory enzymes and HT, as well as apoptosis. Modulation of the intestinal microbiota leading to an increase in beneficial bacteria.
[32]	C57BL/6J mice	MPTP	30 mg/kg	5 days	Intraperitoneal		Chicoric acid	30 and 60 mg/kg	12 days	Oral (gavage)	Neuroprotection against neurotoxicity, partly associated with modification of the gut microbiota, with a reduction in neuroinflammation and gut inflammation.
[33]	C57BL/6J mice	MPTP	20 mg/kg	35 days	Intraperitoneal		Maslinic acid	50 and 100 mg/kg/day	35 days	Oral (gavage)	Prevention of dopaminergic neuron loss, which was achieved by improving motor functions and increasing TH expressions, as well as the levels of DA.
[34]	A53T transgenic C57BL/6 mice						Baichanting compound	363 mg/kg	21 days	Oral	Inhibition of the apoptosis of substantia nigra neurons and alleviation of behavioral disorders.
[35]	C57BL/6 mice	MPTP	30 mg/kg	5 days	Intraperitoneal		Curcumin	25, 100, or 400 mg/kg	28 days	Intragastric	Prevention of the inflammatory effects by the modulation of the gut microbiota.

**Table 3 ijms-25-11629-t003:** Probiotics intervention studies. MPTP: 1-Methyl-4-phenil-1,2,3,6-tetrahydropyridine.

	PD Model					Intervention				
Author	Model	Toxic	Dose	Duration	Administration	Intervention	Dose	Duration	Administration	Major Results
[37]	Male C57BL/6J mice	MPTP	30 mg/kg	4 days	Intraperitoneal	*Lactobacillus plantarum* PS128	(10^9^ CFU/200 μL saline)	28 days	Oral (gavage)	Reduction in glial hyperactivation, oxidative stress, and neuroinflammation and modulation of the dysbiosis of the microbiota.
[39]				7 days	Intraperitoneal	*Lactobacillus plantarum* DP189	(10^9^ CFU/mL)	14 days	Oral	Mitigation of α-Syn accumulation in the SN. Ability to resist oxidative stress injury, inflammatory reaction, and gut microbiota dysbiosis.
[41]				4 days	Intraperitoneal	*Bifidobacterium breve* CCFM1067	(10^9^ CFU/200 μL saline)	39 days	Oral (gavage)	Neuroprotective effects by increasing SCFA levels and decreasing glial hyperactivation, the inflammatory response, and gut microbiota dysregulation.
[38]	Male Wistar rats	6-OHDA	5 mg/mL	4 days	Intracranial	Symprove^TM^ (*Lactobacillus acidophilus* NCIMB 30175, *Lactobacillus plantarum* NCIMB 30173, *Lactobacillus rhamnosus* NCIMB 30174, *Enterococcus faecium* NCIMB 30176)	(10^7^ CFU/day)	24 days	Oral	Reduction in plasma inflammatory markers and SCFAs as well as striatal inflammation by modifying the composition of the microbiota.
[42]	Male C57BL/6J mice	Rotenone	2.5 mg/kg/day	42 days	Intraperitoneal	*Lactobacillus plantarum* PS128	(10^9^ CFU/mL)	42 days	Oral (gavage)	Neuroprotective effects by remodeling the gut microbiota via the miR-155-5p/SOCS1 pathway.
[43]	C57BL/6J mice	Rotenone	30 mg/kg	56 days	Oral	*Lactobacillus plantarum* CCFM405	(10^9^ CFU/200 μL saline)	28 days	Oral (gavage)	Improvement in microbiota composition, reduction in colonic inflammation, and protection of dopaminergic neurons.
[44]	Male C57BL/6J mice	MPTP	30 mg/kg	28 days	Intraperitoneal	*Bifidobacterium animalis subsp. lactis* NJ241	(10^9^ CFU/200 μL saline	28 days	Oral (gavage)	Increase in intestinal GLP-1 levels and activation of nigral PGC-1α signaling for neuroprotection against neuroinflammation.

**Table 4 ijms-25-11629-t004:** Fecal microbiota transplantation intervention studies. MPTP: 1-Methyl-4-phenil-1,2,3,6-tetrahydropyridine.

	PD Model					Intervention			
Author	Model	Toxic	Dose	Duration	Administration	Dose	Duration	Administration	Results
[48]	C57BL/6 mice	Rotenone	30 mg/kg/day	28 days	Oral (gavage)	100 µL/day (100 mg/mL)	14 days	Oral (gavage)	Reversal of the gut microbiota dysbiosis and protection from inflammation mediated by LPS-TLR4 signaling pathway.
[49]		MPTP	20 mg/kg/day	35 days	Intraperitoneal	200 µL/day (10^8^ CFU/mL)	14 days	Intraperitoneal	Reversal of the gut microbial dysfunction and neuroprotection. Reduced activation of microglia and astrocytes in the SN and reduced expression levels of GSK3β, IL-1β, inducible nitric oxide synthase, and phosphorylated PTEN in the SN.
[50]			15 mg/kg	7 days		200 μL/day (10^8^ CFU/mL)	7 days	Oral (gavage)	Augmentation of neurogenesis, improvement in motor function, and restoration of dopaminergic neurons and neurotransmitters.
[51]			30 mg/kg/day	5 days		200 μL/day (1 g/15 mL)	10 days		Regulation of mitochondrial oxidative resistance to neuroinflammation and rescue of nigrostriatal pericyte loss and BBB disruption.

**Table 5 ijms-25-11629-t005:** Other intervention studies. MPTP: 1-Methyl-4-phenil-1,2,3,6-tetrahydropyridine. UCB: Umbilical Cord Blood. UC-MSCs: Umbilical Cord-Mesenchymal Stem Cells. SNpc: Substantia Nigra pars compacta. GLP-1: Glucagon-like peptide-1. MAO-B: Monoamine oxidase-B. DA: Dopamine.

	PD Model					Intervention			
Author	Model	Toxic	Dose	Duration	Administration	Dose	Duration	Administration	Results
[52]	Sprague–Dawley rats	MPTP	20 mg/kg/day	15 days	Intraperitoneal	UCB (500 µL)	5, 10, and 12 days	Intravascular	Improvement in motor function, gut motility, and dopaminergic neuronal survival. Reduction in proinflammatory cytokines in both the SNpc and the intestinal mucosa and dampened inflammation-associated gut microbiota.
[53]	C57BL/6 mice		30 mg/kg/day	5 days		UC-MSCs 1 × 10^6^ cells/40 μL	7 days	Intranasal	Improvement in motor dysfunction and repair of the degeneration of dopaminergic neurons by inhibiting activated glial cells, decreasing proinflammatory cytokine release, maintaining normal mucosal barrier, and restricting NF-кB expression.
[54]			20 mg/kg/day	7 days		Ceftriaxone 200 mg/kg	7 days	Intraperitoneal	Regulation of the intestinal microbiota. Relief of motor dysfunction and decreased exploratory ability. Increased intestinal barrier integrity, reduced inflammation of the colon and brain, and neuroprotective effect.
						FMT 200 μL (10^7^ CFU/mL)		Oral (gavage)	
[55]			30 mg/kg/day			Dioscin 20, 40, and 80 mg/kg	28 days		Remodeling of the gut microbiota and regulation of bile acid-mediated oxidative stress and neuroinflammation by targeting GLP-1 signaling.
[56]			20 mg/kg			Vancomycin 100 mg/kg/day	14 days		Suppression of MAO-B expression and DA catabolism possibly via the gut and brain TLR4/MyD88/NF-κB/TNF-α.

**Table 6 ijms-25-11629-t006:** Human intervention studies.

Authors	Trial	Population	Administration	Duration	Intervention	Results
[84]	Double-blinded, placebo-controlled, randomized clinical trial	n: 45 (25 M/20 W) 35–75 yr (median age 66 yr) n: 30 Mild to moderate Parkinson’s disease + dysbiotic fecal microbiota	Colonic single-dose anaerobically prepared FMT. FMT via colonoscopy	48 weeks	Single-dose FMT via colonoscopy (n: 30) Single-dose placebo via colonoscopy (n: 15)	FMT was linked to mostly transient gastrointestinal adverse events without clinically meaningful improvements. Stronger increase in dopaminergic medication and improvement in certain motor and non-motor outcomes in the placebo group.
[81]	12-week randomized, double-blinded, placebo-controlled clinical trial	n: 60 (50–90 yr) Parkinson’s disease	Probiotic administration via capsule ingestion	12 weeks	Capsule: 8 × 10^9^ CFU/day probiotic (*Lactobacillus acidophilus*, *Bifidobacterium* *bifidum*, *Lactobacillus reuteri*, and *Lactobacillus fermentum*, each 2 × 10^9^ CFU/g) (n: 30; 67.7 yr) Placebo (n: 30; 68.2 yr)	Decreased MDS-UPDRS. Reduced high-sensitivity C-reactive protein and malondialdehyde and enhanced glutathione levels. Reduction in insulin levels and insulin resistance.
[80]	Single-arm, Mediterranean diet intervention study (PREDIMED trial protocol)	n: 8 Parkinson’s disease	Mediterranean diet	5 weeks	Mediterranean diet intervention. Olive oil; ≥2 daily servings of vegetables; ≥2–3 daily serving of fresh fruits; ≥3 weekly servings of legumes; ≥3 weekly servings of fish or seafood; ≥3 weekly servings of nuts or seeds; white meats; and cook at least twice a week with a sofrito sauce. *Ad libitum*: nuts, eggs, fish, seafood, low-fat cheese, and whole-grain cereals.	Constipation syndrome scores decreased. Modified gut microbiota: Bilophila slightly decreased, Roseburia significantly lowered. No differences in markers of intestinal permeability.
[82]	Double-blinded on the day of the procedure and unblinded at the time of data analysis	n: 6 3 M, 47–73 yr, median age 52 yr 3 W, 52–71 yr, median age 64 yr Parkinson’s disease and constipation	FMT infused via colonoscopy	24 weeks	300 mL of fecal suspension of donor stool delivered in three portions: 100 mL at the terminal ileum, 100 mL at the cecum, and 100 mL along the rest of the colon	Improvement in PD motor and non-motor symptoms, including constipation, at 6 months.
[83]	Randomized, placebo-controlled trial	n: 54 30–85 yr Mild to moderate Parkinson’s disease	Fecal microbiota transplantation via capsule ingestion	12 weeks	FMT capsule (n: 27; 15 M, 12 W, median age 60.5 yr) Placebo (n: 27; 17 M, 10 W, median age 62.6 yr)	Significant improvement in PD-related autonomic symptoms (MDS-UPDRS total score). Improved gastrointestinal disorders and a marked increase in the complexity of the microecological system.

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
