# Peer review of "Convergence of Neuroinflammation, Microbiota, and Parkinson’s Disease: Therapeutic Insights and Prospects"

_ijms, 2024, doi:10.3390/ijms252111629_

Round 1
Reviewer 1 Report
Comments and Suggestions for Authors
In the manuscript # ijms-3145659 by Domínguez Rojo, “Convergence of Neuroinflammation, Microbiota, and Parkinson's Disease: Therapeutic Insights and Prospects” discuss the putative connections between neuroinflammatory processes and intestinal microbiota alterations in the progression of Parkinson's Disease pathology. They aim to make the connection between alterations in gut microbiota composition and Parkinson's Disease, suggesting a potential modulatory role in disease progression. They suggest that a bidirectional communication along the gut-brain axis has become important in comprehending the pathogenesis of PD. Furthermore, the authors aim to explore emerging therapeutic strategies that target these interconnected pathways, providing insights into potential avenues for Parkinson's Disease treatment. This is a rather speculative delineation of a new diseases concept that require a solid scientific foundations.
Improvement suggestions:
1.)
Please provide in the beginning of the manuscript a paragraph clearly outlining the mechanistic-principles of the gut microbiota-brain communication axis, with the strong accent on the immune-modulation.
2.)
Please support your claims with the relevant clinical studies in humans. Tables 1-4 all summarize rodent-based experimental data. Laboratory rodent are feed with standardized diet and their gut-microbiota has little to do with the rodents’ gut microbiota in the wild. Please provide a tabular summary of relevant studies in humans.
Comments on the Quality of English Language
The anguage is in general OK. In isolated instances Spanish-language phrases have been translated into English.
Author Response
In the manuscript # ijms-3145659 by Domínguez Rojo, “Convergence of Neuroinflammation, Microbiota, and Parkinson's Disease: Therapeutic Insights and Prospects” discuss the putative connections between neuroinflammatory processes and intestinal microbiota alterations in the progression of Parkinson's Disease pathology. They aim to make the connection between alterations in gut microbiota composition and Parkinson's Disease, suggesting a potential modulatory role in disease progression. They suggest that a bidirectional communication along the gut-brain axis has become important in comprehending the pathogenesis of PD. Furthermore, the authors aim to explore emerging therapeutic strategies that target these interconnected pathways, providing insights into potential avenues for Parkinson's Disease treatment. This is a rather speculative delineation of a new diseases concept that require a solid scientific foundations.
Response: We greatly appreciate the reviewer's comments to improve the quality of the article, and have therefore taken the specific suggestions into account and modified the article accordingly.
Improvement suggestions:
1.)
Please provide in the beginning of the manuscript a paragraph clearly outlining the mechanistic-principles of the gut microbiota-brain communication axis, with the strong accent on the immune-modulation.
Response: Following the reviewer's recommendations, we have added a paragraph in the section 1.3. Interconnections and Emerging Insights (lines 58-70) to describe the main mechanisms of the gut microbiota-brain communication axis.
2.)
Please support your claims with the relevant clinical studies in humans. Tables 1-4 all summarize rodent-based experimental data. Laboratory rodent are feed with standardized diet and their gut-microbiota has little to do with the rodents’ gut microbiota in the wild. Please provide a tabular summary of relevant studies in humans.
Response: Following the reviewer's recommendations, we have added a table (table 6) with the interventions described in humans. In addition to this table, a section (Human interventions) has been added to the manuscript (lines 518-546).
Reviewer 2 Report
Comments and Suggestions for Authors
Dear authors,
Interplay between gut and brain possibly play an important role in brain disorders, including Parkinson’s disease (PD). Emerging data has indicated that microbiota and metabolite can elicit neuroinflammation and contribute to PD pathogenesis. Intervention on microbiota is a potential strategy for PD treatment. Your manuscript well summarized investigations on this topic, but most results are observational changes, no direct interplay mechanism are proposed. It is unclear how microbiota changes activate neuroinflammation in your manuscript which is a key message audience may want to see.
Since most data are from drug induced rodent PD models, and those models only can partially recapitulate PD pathomechanism, I suggest you include compartment data in PD patients to support the investigation on the rodent models. The audience would like to see insightful views in the discussion rather than general description. Possible molecular mechanism needs to proposed in the discussion or perspective. The hallmark pathology of PD, alpha-synuclein aggregation, Lewy neurite/body should not be missed in discussion.
There are some details need to adjust/correct”
1. In figure 1, the author used mid brain picture of human while a mouse image centered, it is not compatible.
2. I think it is inappropriate to put a human digestive track image in Figure 2, Since the manuscript mostly talked about mouse model researches.
3. It would be better to incorporate Figure 1, 2 into main text, not leave them at the end separately.
4. For tables, it would be better to present major result of each research.
5. In line 114, “system nervous system” need to correct.
6. In line 173, “In another study, (23), a rotenone-induced disease”, the “,” before “(23)” need to delete.
7. In table 3, there are 2 titles list as intervention, need to adjust.
Thank you for sharing!
Best!
Author Response
Reviewer 2
Dear authors,
Interplay between gut and brain possibly play an important role in brain disorders, including Parkinson’s disease (PD). Emerging data has indicated that microbiota and metabolite can elicit neuroinflammation and contribute to PD pathogenesis. Intervention on microbiota is a potential strategy for PD treatment. Your manuscript well summarized investigations on this topic, but most results are observational changes, no direct interplay mechanism are proposed. It is unclear how microbiota changes activate neuroinflammation in your manuscript which is a key message audience may want to see.
Since most data are from drug induced rodent PD models, and those models only can partially recapitulate PD pathomechanism, I suggest you include compartment data in PD patients to support the investigation on the rodent models. The audience would like to see insightful views in the discussion rather than general description. Possible molecular mechanism needs to proposed in the discussion or perspective. The hallmark pathology of PD, alpha-synuclein aggregation, Lewy neurite/body should not be missed in discussion.
Response: We appreciate the comments of reviewer 2 to improve the quality of the article. In this regard, and in line with what reviewer 1 also suggested, we have added a table (table 6) and a section (section 3, Human interventions) with the studies carried out in humans.
As to the possible mechanism, there are no data available to date, except for those already mentioned in the manuscript. In this regard, we have added a paragraph in the section 1.3. Interconnections and Emerging Insights (lines 58-70) to describe the main mechanisms of the gut microbiota-brain communication axis.
There are some details need to adjust/correct”
- In figure 1, the author used mid brain picture of human while a mouse image centered, it is not compatible.
Response: Sorry for the confusion, the human mid brain has been replaced by a murine brain in the figure1.
- I think it is inappropriate to put a human digestive track image in Figure 2, Since the manuscript mostly talked about mouse model researches.
Response: Sorry again for the confusion, the human digestive track image in Figure 2 has been replaced by a murine digestive track image
- It would be better to incorporate Figure 1, 2 into main text, not leave them at the end separately.
Response: Following the reviewer's recommendation, we have inserted figure 1 and 2 in the main text.
- For tables, it would be better to present major result of each research.
Response: We appreciate the reviewer's comment and following his indications, we have added a column with the results in the tables.
- In line 114, “system nervous system” need to correct.
Response: We have corrected this error in the manuscript.
- In line 173, “In another study, (23), a rotenone-induced disease”, the “,” before “(23)” need to delete.
Response: We have removed the comma.
- In table 3, there are 2 titles list as intervention, need to adjust.
Response: Thank you for your comment. We have added the term “Compound” instead of intervention in one title.
Reviewer 3 Report
Comments and Suggestions for Authors
The work proposed by the authors highlights the importance of the gut microbiota in the development of neuroinflammation and a possible progression to Parkinson's disease. Additionally, the authors thoroughly present current therapeutic approaches to minimize the effects of neuroinflammation caused by the microbiota. Overall, the work is well-structured and has good scientific quality. I highlight a few minor changes to be made.
Lines 71-73. One of the most evident hallmarks of Parkinson's disease is the aggregation of alpha-synuclein, which is also observed in the induction by Proteus mirabilis. I believe it would be interesting to add this information related to the research by Huh E, Choi (reference 10).
Tables: I would recommend the authors standardize the duration times of the inductions of the models/interventions performed. The tables use days, weeks, or months. I would recommend changing all to days, as it would facilitate comparison.
I believe it would be interesting to explain some acronyms used, such as: NO (Line 159), 6S and GE (Line 184).
Line 262: I believe there is an extra word ("que") in the text.
Line 377: The topic “Umbilical Cord Blood Plasma Therapy” does not clearly relate this intervention to the gut microbiota.
Author Response
Reviewer 3
The work proposed by the authors highlights the importance of the gut microbiota in the development of neuroinflammation and a possible progression to Parkinson's disease. Additionally, the authors thoroughly present current therapeutic approaches to minimize the effects of neuroinflammation caused by the microbiota. Overall, the work is well-structured and has good scientific quality. I highlight a few minor changes to be made.
Response: Thank you for your comment.
Lines 71-73. One of the most evident hallmarks of Parkinson's disease is the aggregation of alpha-synuclein, which is also observed in the induction by Proteus mirabilis. I believe it would be interesting to add this information related to the research by Huh E, Choi (reference 10).
Response: Thanks for the suggestion. it has been added to the manuscript.
Tables: I would recommend the authors standardize the duration times of the inductions of the models/interventions performed. The tables use days, weeks, or months. I would recommend changing all to days, as it would facilitate comparison.
Response: Following the reviewer's recommendations, we have standardize the duration times to days
I believe it would be interesting to explain some acronyms used, such as: NO (Line 159), 6S and GE (Line 184).
Response: We have explain the acronyms
Line 262: I believe there is an extra word ("que") in the text.
Response: we have deleted it
Line 377: The topic “Umbilical Cord Blood Plasma Therapy” does not clearly relate this intervention to the gut microbiota.
Response: We apologize to the reviewer for not agreeing with his suggestion, we understand that the aforementioned intervention can alter the composition of the microbiota, as shown in the table 4 in the relevant results section.
Round 2
Reviewer 2 Report
Comments and Suggestions for Authors
All concerns I raised in last round reviews have been well addressed, however, in tables, descriptions of major findings are redundant, I suggest a concise and brief rewriting. For figure 1, the mouse brain section is not a typical plane to show nigra, and TH positive cells (black dot) are not in nigra. I do think the picture need to correct.
Author Response
Comments 1: All concerns I raised in last round reviews have been well addressed, however, in tables, descriptions of major findings are redundant, I suggest a concise and brief rewriting. For figure 1, the mouse brain section is not a typical plane to show nigra, and TH positive cells (black dot) are not in nigra. I do think the picture need to correct.
Response: We encourage you to contribute your insights and recommendations. In response to your feedback, we have streamlined the main results presented in the tables and corrected the figure 1